# A Comparison of Brain-State Dynamics across Common Anesthetic Agents in Male Sprague-Dawley Rats

**DOI:** 10.3390/ijms23073608

**Published:** 2022-03-25

**Authors:** Rachel Ward-Flanagan, Alto S. Lo, Elizabeth A. Clement, Clayton T. Dickson

**Affiliations:** 1Neuroscience and Mental Health Institute, University of Alberta, Edmonton, AB T6G 2E1, Canada; wardflan@ualberta.ca (R.W.-F.); eaclemen@ualberta.ca (E.A.C.); 2Department of Psychology, University of Alberta, Edmonton, AB T6G 2R3, Canada; aslo@ualberta.ca; 3Department of Physiology, University of Alberta, Edmonton, AB T6G 2H7, Canada; 4Department of Anesthesiology and Pain Medicine, University of Alberta, Edmonton, AB T6G 2G3, Canada

**Keywords:** urethane, chloral hydrate, ketamine-xylazine, pentobarbital, propofol, isoflurane, sleep, unconsciousness, EEG

## Abstract

Anesthesia is a powerful tool in neuroscientific research, especially in sleep research where it has the experimental advantage of allowing surgical interventions that are ethically problematic in natural sleep. Yet, while it is well documented that different anesthetic agents produce a variety of brain states, and consequently have differential effects on a multitude of neurophysiological factors, these outcomes vary based on dosages, the animal species used, and the pharmacological mechanisms specific to each anesthetic agent. Thus, our aim was to conduct a controlled comparison of spontaneous electrophysiological dynamics at a surgical plane of anesthesia under six common research anesthetics using a ubiquitous animal model, the Sprague-Dawley rat. From this direct comparison, we also evaluated which anesthetic agents may serve as pharmacological proxies for the electrophysiological features and dynamics of unconscious states such as sleep and coma. We found that at a surgical plane, pentobarbital, isoflurane and propofol all produced a continuous pattern of burst-suppression activity, which is a neurophysiological state characteristically observed during coma. In contrast, ketamine-xylazine produced synchronized, slow-oscillatory activity, similar to that observed during slow-wave sleep. Notably, both urethane and chloral hydrate produced the spontaneous, cyclical alternations between forebrain activation (REM-like) and deactivation (non-REM-like) that are similar to those observed during natural sleep. Thus, choice of anesthesia, in conjunction with continuous brain state monitoring, are critical considerations in order to avoid brain-state confounds when conducting neurophysiological experiments.

## 1. Introduction

Anesthesia is one of the most common tools used in neuroscientific research. Its prevalence is unsurprising given the exceptional experimental control that it affords, granting researchers the ability to perform surgical manipulations that would otherwise be technically or ethically impossible [1]. Yet, despite the predominance of anesthesia in animal research, the appropriate choice of an anesthetic agent can be difficult, as neurophysiological effects vary by anesthetic agent [2,3], dosage [4,5], animal model [6], and pharmacological mechanisms [7,8,9]. Furthermore, while the agent-specific brain states produced by anesthetics can significantly impact experimental outcomes [10], often the choice of anesthetic agent is not rationalized, and brain state goes unreported in many neurophysiological studies using anesthesia.

Consequently, our aim was to conduct a controlled comparison of the ongoing electrographic activity evoked at a surgical plane of anesthesia under six common research anesthetics using a ubiquitous research model, the Sprague-Dawley rat. Though at clinical doses, most anesthetics act on multiple molecular targets, for the sake of brevity we mainly focused on the primary pharmacological mechanisms of our chosen agents. In order for our comparison to serve as a preliminary guide for researchers in choosing a suitable anesthetic, we tested easily accessible, commonly used anesthetics with a variety of primary pharmacological actions, including: ketamine-xylazine (KET-XYL), which acts as an n-methyl-D-aspartate (NMDA) receptor antagonist [11] and an alpha-2 adrenergic agonist [12], respectively; pentobarbital (PTB), isoflurane (ISO), propofol (PRO), and chloral hydrate (CH), which are all reported to share a primary pharmacological mechanism, i.e., potentiating γ-aminobutyric acid (GABA)-ergic activity [13,14,15,16,17]. We also tested urethane (ethyl carbamate), which potentiates a resting potassium conductance in order to decrease membrane input resistance, thus hyperpolarizing excitatory neocortical pyramidal neurons as its primary pharmacological mechanism [18,19].

We also reasoned that our direct comparison would serve to determine which, if any, anesthetic agents tested could act as pharmacological proxies for altered states of unconsciousness, such as sleep and coma [20]. Previous work from our research group using rodents has established that urethane anesthesia produces electroencephalographic (EEG) activity characterized by spontaneous, cyclical alternations between: (1) a state of forebrain activation (i.e., low-voltage fast activity in the cortex, concurrent with theta activity in the hippocampus; ~4 Hz), and (2) a deactivated state characterized by global, synchronous, slow-oscillatory activity (~1 Hz; [21,22]). These brain-state dynamics closely resemble the spontaneous EEG fluctuations between rapid-eye movement (REM) and non-REM (nREM) in natural sleep in terms of periodicity, duration, and concomitant changes in physiology such as heart rate, breathing rate and temperature [23,24,25]. Due to these previous findings, and the unusual primary pharmacological mechanism of urethane, we hypothesized that when compared to other commonly used anesthetics, only urethane anesthesia would serve as a viable pharmacological proxy for the full neurophysiological dynamics of natural sleep.

Using identical recording conditions to enable direct comparisons, we monitored ongoing brain-state dynamics via intracranial electrodes placed in the neocortex and hippocampus, as well as any associated changes in the plane of anesthesia either following bolus doses of the anesthetic, or conversely during subsequent metabolism of the anesthetic agent. We evaluated consistencies and divergences in brain activity across our tested anesthetic agents at a surgical plane of anesthesia, using spectral analysis of the EEG recordings. Our comparison revealed that three different states could be observed: coma-like burst-suppression (PTB, ISO, PRO), nREM-like slow-oscillatory activity (KET-XYL), and sleep-like cyclical oscillations between an REM-like and nREM-like state (urethane, CH).

## 2. Results

### 2.1. Urethane

As with our previous studies [21,22], the rats in the urethane-anesthetized group exhibited robust cyclical and spontaneous alternations of forebrain state during long-term neocortical and hippocampal local-field-potential recordings (Figure 1A,B). Two electrographically distinct states were observed: an activated pattern, consisting of low-voltage fast activity in the neocortex, concomitant with rhythmic theta activity (3.90 ± 0.13 Hz) in the hippocampus; and a deactivated state characterized by large amplitude, slow-oscillatory (~1 Hz: CTX: 1.18 ± 0.07 Hz; HPC: 0.96 ± 0.09 Hz) activity in both the neocortex and the hippocampus (Figure 1C). The average period for these highly rhythmic state alternations was 11.6 ± 1.0 min (*n* = 10).

Furthermore, additional bolus doses of urethane did not change the rhythmicity nor the periodicity of state changes, although they did change the proportion of time spent in the two states. In a subset of urethane-anesthetized rats, supplemental doses of urethane (0.52 ± 0.14 g/kg; *n* = 4) decreased the proportion of time spent in the activated state (42.2 ± 12.0% per cycle; Figure 1D). However, in the subset of rats that were not administered supplemental doses of urethane, the gradual metabolism of urethane (over an average time period of 2.46 ± 0.28 h; *n* = 6) increased the proportion of time spent in the activated state (29.83 ± 6.62% per cycle; Figure 1E).

During these long-term recordings, the anesthetic plane was monitored via a nociceptive stimulus to the hind paw. However, consistent with our previous observations [21], no reflexive withdrawal response was observed in any of the urethane-anesthetized animals regardless of brain state, indicating that the observed changes in brain state were not due to a decline in the anesthetic plane.

### 2.2. Ketamine-Xylazine

In accordance with our previous research [26], KET-XYL anesthesia was found to evoke a particularly stable and unitary state of large amplitude, slow-oscillatory activity in both the neocortex and hippocampus (Figure 2A,B) at an average peak frequency of 1.6 ± 0.07 Hz (*n* = 5; Figure 2C). We also observed some slow-oscillation-coupled beta activity in the cortex with a wide bandwidth range between 10–25 Hz (the 10 Hz peak in 2C is an example of such activity). Over a 40 min period of metabolism, there was a gradual decrease in the amplitude of slow-oscillatory local-field potentials in the raw neocortical and hippocampal recordings, which could also be observed in the power spectra in the 0.5–2 Hz bandwidth (Figure 2D). While this state of synchronized slow-oscillatory activity resembled the activity observed during the deactivated state under urethane, the KET-XYL-anesthetized rats did not exhibit any cyclical alternations in brain state while at a surgical plane.

Supplemental i.v. bolus doses of KET-XYL (16.6 ± 1.9 mg/kg and 1.5 ± 0.1 mg/kg; *n* = 5), consistently and rapidly increased the amplitude of the slow-oscillatory activity in the neocortex and hippocampus (Figure 2E,F). In the spectral analyses, these increases were observed in the power of the slow-oscillatory signal (~1 Hz) and gamma bandwidth (30–40 Hz; Figure 2F). The differential pharmacodynamic and pharmacokinetic profiles of ketamine and xylazine likely have a role in the brain-state effects we observed following the bolus doses and metabolism of the drugs, as each drug would exert a more powerful influence dependent on the time from administration.

On the other hand, when animals were left to metabolize the drug over long periods (>40 min), a shift in both brain and physiological state (i.e., increased respiration rate) could suddenly occur. The brain state during this time period exhibited patterns of neocortical low-voltage fast activity, and hippocampal theta similar to the activity observed in the activated state in the urethane-anesthetized rats and during REM sleep. However, in contrast with the stable anesthetic plane produced by urethane, these electrophysiological changes in KET-XYL coincided with a reflexive withdrawal to a hind-paw pinch. This key difference indicates that the observed shifts in the EEG state were reflective of a loss of a surgical plane, rather than a neurophysiological state analogous to REM in natural sleep or the activated state under urethane. Accordingly, when animals exhibited these shifts, they were promptly administered a supplementary dose of KET-XYL to restore a surgical plane of anesthesia. This promptly restored the slow-wave pattern that was characteristic of the baseline anesthetized recordings.

### 2.3. Pentobarbital

Burst-suppression activity was consistently observed at a surgical plane in both the neocortex and hippocampus of the PTB-anesthetized rats (Figure 3A,B). Burst-suppression is a pattern of activity characterized by isoelectricity interspersed with high-amplitude spikes and typically serves as an indicator of very deep levels of anesthesia [27,28]. It is also associated with brain activity observed in patients with brain damage or coma [5].

Across neocortical and hippocampal sites, bursts occurred with a high degree of coincidence (Figure 3C). However, during periods of isoelectricity in the neocortex, the hippocampus showed low-voltage, faster activity with a peak amplitude in the 11–18 Hz bandwidth (Figure 3C). Over 40 min periods of recording, without supplemental infusions of PTB, we observed a near-linear increase in electrographic amplitude in both the neocortex and the hippocampus (Figure 3A–D). Correspondent with these increases in amplitude was a gradual return of spectral power for all frequencies, with a preferential increase in lower-frequency bandwidths (0–10 Hz; Figure 3D).

A subset of this group of rats was administered bolus increases of PTB. At baseline, in the pre-bolus phase, we observed an average isoelectric period between bursts of 1.1 ± 0.1 s (*n* = 3), during a stable surgical plane of anesthesia (Figure 3E,F). Subsequently, when a bolus dose of PTB (6.5 mg) was administered, the duration of the isoelectric periods between bursts significantly increased on average by 6.8 ± 2.8 s (*n* = 3; F_(1,503)_ = 163.5; *p* < 0.0001; Figure 3F). The concurrent periods of isoelectricity in the cortex and the low-voltage fast activity in the hippocampus that was observed post-bolus infusion translated to a decrease in spectral power across all frequencies, with lower frequencies (0.5–4.5 Hz) being preferentially depressed (Figure 3G).

When the rats were allowed to metabolize the bolus dose without additional infusions, we observed a reciprocal and significant change in the isoelectric inter-burst interval; an average decrease of 6.2 ± 3.2 s over 40 min (*n* = 3; F_(1,459)_ = 183.9; *p* < 0.0001. While spectral power (and signal amplitude) gradually recovered to pre-infusion levels, there were no dramatic shifts in brain state, nor in the anesthetic plane.

### 2.4. Isoflurane

At a surgical plane of anesthesia, ISO (2.0% in medical oxygen) also generated a burst-suppression pattern of electrographic activity in the forebrain, with remarkable similarities to the pattern of burst-suppression activity observed under PTB (Figure 4A,B). As with PTB, bursts tended to occur concomitantly at both the neocortical and hippocampal sites, while periods of isoelectricity in cortex corresponded to low-voltage fast activity (8–15 Hz; *n* = 5) in the hippocampus (Figure 4C).

We also recorded EEG activity under reduced concentrations while closely monitoring anesthetic state. At a concentration of 1.5% ISO, the average inter-burst interval was 2.9 ± 1.7 s (*n* = 3; Figure 4D–F). Subsequent changes from this to a higher concentration (2%) significantly increased the duration of the isoelectric period between bursts, which corresponded to an average increase of 24.4 ± 14.6 s (F_(1,263)_ = 96.11; *p* < 0.0001; Figure 4D). Interestingly, at further decreased concentrations of ISO (1%), we observed changes in brain state in three of the five rats tested. Unlike the effects observed at 1.5%, these changes in brain state were marked by a shift in neocortical activity from burst-suppression to low-voltage, fast irregular activity, and the appearance of theta activity (3–12 Hz) in the hippocampus. While the state produced by this transition was reminiscent of the activated state observed in urethane anesthesia and REM sleep, it was also accompanied by a reflexive withdrawal to a hind-paw pinch, indicating a loss of the surgical plane. This was immediately rectified with an increase in the delivered concentration of ISO to 2% in order to restore an appropriate level of anesthesia, which was accompanied by a broadband decrease in power (Figure 4E). Therefore, our results show that at surgical planes of anesthesia, the only form of activity apparent in the forebrain was burst suppression, and any observed changes in state-dependent activity were reflective of a loss of a surgical plane.

### 2.5. Propofol

Due to a relatively short duration of anesthetic effect, PRO is often administered via continuous infusion in order to ensure a stable plane of anesthesia [29]. Thus, we investigated the neurophysiological state evoked by PRO at a steady rate of infusion and assessed differences in the circulating levels of PRO by administering supplementary bolus doses, and/or by temporarily discontinuing continuous infusion until a change in brain state (or anesthetic plane) was observed.

At a continuous infusion rate of 60 mg·kg^−1^·h^−1^, PRO induced a stable pattern of burst-suppression activity (Figure 5A–C), consistent with previous reports [30], which showed amplified power in the 0.5–2 Hz bandwidth (Figure 5C,D). This activity was similar to the burst-suppression activity observed in both PTB and ISO, albeit with a shorter average period of isoelectricity during the inter-burst intervals (0.83 ± 0.06 s; *n* = 3). A bolus infusion (2 mg) of PRO produced a broadband decrease in power (Figure 5E,F) accompanied by a significant increase in the average period of isoelectric activity between bursts of 1.1 ± 0.4 s (F_(1,574)_ = 136.0; *p* < 0.0001; *n* = 3; Figure 5G). This observed increase in the inter-burst interval was analogous to the increases in isoelectric activity observed in both ISO and PTB following supplemental doses of anesthetic.

The temporary suspension of the continuous infusion of PRO (over 10 min) coincided with a gradual increase in broadband power, with a preferential increase in the delta-frequency bandwidth (0-3 Hz) in both the neocortex and the hippocampus (Figure 5H,I). Additionally, over these 10 min, the duration of isoelectric inter-burst intervals significantly decreased by an average of 1.1 ± 0.2 s (*n* = 3; F_(1,562)_ = 87.6; *p* < 0.0001). However, this transition coincided with a reflexive withdrawal of the hind paw to a nociceptive stimulus indicating a loss of the surgical plane, and was subsequently rectified by returning the animal to continuous infusion of PRO. Hence, PRO, PTB and ISO all appear to evoke a similar burst-suppression pattern of altered brain activity during surgical planes of anesthesia.

### 2.6. Chloral Hydrate

Similar to the PRO group, we administered CH at a constant rate of infusion (150 mg·kg^−1^·h^−1^) following an initial bolus dose of 200 mg/kg. Perhaps surprisingly, especially based on its similar pharmacological profile to PTB, ISO and PRO, the CH-anesthetized rats exhibited spontaneous and cyclically occurring alternations between two distinct forebrain states, remarkably like those observed during urethane anesthesia (Figure 6A–C and Figure 1A–C). We observed both a state of forebrain activation that consisted of low-voltage, fast activity in the cortex, coinciding with a prominent theta (3.30 ± 0.11 Hz) oscillation in the hippocampus, alternating with a deactivated state consisting of large-amplitude, slow-oscillatory activity in both the cortex (0.73 ± 0.07 Hz) and hippocampus (0.64 ± 0.03 Hz; Figure 6B,C). The average period of these cyclical alternations was also highly similar to the period observed in urethane, at 10.12 ± 0.58 min per cycle.

Further overlaps with urethane were observed when a subset of rats was given supplemental bolus doses of CH in addition to the continuous infusion (total supplemental dose: 50.0 ± 5.0 mg, *n* = 3; Figure 6D). During the first two 15 mg i.v. bolus infusions, the proportion of time spent in the activated state decreased by an average of 16.65 ± 4.14% per cycle, with an inverse increase in the proportion of time spent in the deactivated state (Figure 6D). We did not include the change in proportion of time spent in the activated state from the second to third 15 mg bolus in our calculations, since two of the three animals reached a ceiling after the second bolus infusion, spending 100% of the cycle in the deactivated state, meaning that no change was observed from the second to third bolus infusion.

When the continuous infusion of CH was discontinued in the same rats (Figure 6E), we observed an average increase of 12.47 ± 3.68% in the proportion of time spent in the activated state per cycle, once a distinguishable change in the proportion of time spent in the activated state was observed. Importantly, neither the additional doses of CH nor its metabolism altered the period length of each cycle (Figure 6E).

In a final parallel to urethane anesthesia, these electrophysiological dynamics do not appear to be due to a lessening of the anesthetic state. During baseline recordings, all rats in the CH condition were receiving a continuous infusion of the drug, and we also observed no reflexive withdrawal response to a hind-paw pinch, irrespective of brain state.

## 3. Discussion

Under identical recording conditions, each of the six tested anesthetic agents produced one of three distinct patterns of EEG activity at a surgical plane of anesthesia:
Burst-suppression, which is a brain state characterized by short periods of high-amplitude, high-frequency bursts interspersed with longer periods of isoelectric activity, and is typically associated with brain damage, hypothermia or coma [5]. This pattern of EEG activity was observed during PTB [31], ISO [32], and PRO [30] anesthesia.A unitary state of synchronized, slow-oscillatory activity similar to the rhythmic on-off (up/down) field fluctuations observed during nREM sleep [33] was observed during KET-XYL anesthesia.Spontaneous, cyclical alternations between activated and deactivated brain states analogous to the REM/nREM cycle during natural sleep, in terms of both electrographic features and dynamics, was observed during both urethane [21] and CH anesthesia.


The agent-specific diversity of these brain states highlights how crucial the appropriate choice of anesthetic is within an experimental paradigm, and indeed the necessity to report brain state in order to accurately interpret in vivo neurophysiological experimental outcomes [10].

It is important to note that any major deviations of forebrain activity towards more activated patterns, from either the patterns of burst suppression produced by PTB, ISO, and PRO, or from the slow-oscillatory activity observed during KET-XYL following periods of metabolism (or in the case of ISO, a lower gaseous concentration) were associated with the loss of a surgical plane of anesthesia. In these cases, the loss of a clinical plane of anesthesia was indicative of an imminent return to consciousness. Consequently, a continuous observation of brain state not only contextualizes neurophysiological experimental outcomes, but also serves as an online index of the depth of anesthesia when agent-specific EEG signatures are appropriately monitored [2].

In contrast to the metabolically coupled changes in brain state observed in PTB, ISO, PRO and KET-XYL, we have previously demonstrated that brain-state alternations under urethane anesthesia are not associated with a lessening of the anesthetic plane [21], nor are they significantly altered over long recording periods [34]. Here, we further demonstrated, not only in urethane, but also in CH, that no reflexive withdrawal to a hind-paw pinch was observed in either the activated or the deactivated brain state, and furthermore, that bolus infusions or metabolism of urethane or CH did not alter the periodicity of brain-state alternations, only the proportion of time spent in either the activated or deactivated state per cycle. These data indicate that the alternations between dichotomous brain states under the two anesthetics were not attributable to changes in the anesthetic plane. In this respect, both urethane and continuous i.v. administration of CH provide extremely stable and tractable experimental models for the alternating electrophysiological features and dynamics of sleep, while also allowing for surgical manipulations that might not be technically or ethically feasible in naturally sleeping animals.

### 3.1. Anesthesia as a Model for the Brain States Associated with Altered States of Consciousness

The dose-dependent and agent-specific control of brain state that is afforded by anesthesia provides an unparalleled pharmacological analog to mimic the brain states observed in many altered states of consciousness, such as specific components of natural sleep, and coma [20,26]. As we demonstrate here, KET-XYL has applications for modelling the rhythmic large-amplitude, slow-oscillatory activity, which is archetypical of nREM sleep. KET-XYL also provides the added experimental advantage of brain-state stationarity, since the highly transient nature of natural sleep makes the cohesive analysis of a single brain state technically challenging [26]. Accordingly, the stability of the forebrain slow oscillation under KET-XYL provides a useful means to explore aspects of nREM such as the hippocampal slow oscillation [22], and intracellular thalamocortical dynamics [33].

Anesthesia is also an excellent tool for replicating brain states associated with coma. This is perhaps best evidenced by its clinical use for inducing medical coma in order to manage conditions such as refractory status epilepticus and in treating traumatic brain injuries [35,36]. Our data show that surgical-plane levels of PTB, ISO, and PRO all provide effective models to probe the mechanistic complexities of the coma-like brain state of burst suppression, such as cortical hyperexcitability [4], neurovascular coupling [37], and local cortical spatiotemporal dynamics [38]. Indeed, ISO was recently shown to exhibit the same dysregulation in homeostatic neural-firing rates as nREM in pre-symptomatic mouse models of familial Alzheimer’s disease [39], indicating that ISO may serve as a useful model to explore the connection between the subclinical epileptiform activity and network hyperexcitability that is observed in some models of Alzheimer’s disease [40,41].

However, it is important for researchers choosing any of these anesthetics to be aware that PTB, ISO and PRO produce agent-specific differences in both the burst-suppression architecture and the duration of suppression [42,43], which may arise from differential pharmacological mechanisms, such as distinct binding sites on GABA-A receptors [7]. Likewise, KET-XYL has its own experimental caveats, as it has been reported to elicit highly elevated power in the gamma bandwidth (30–100 Hz; [44]), which we also observed when we administered supplementary bolus doses of KET-XYL. Consequently, further considerations may need to be taken into account based on agent-specific neurophysiological characteristics, and findings in these models would likely need to be replicated in the endogenously occurring altered state of consciousness. In addition, researchers employing any of these anesthetics to model specific brain states would need to continuously monitor brain state [10]. Nonetheless, when the limitations of a pharmacological agent are accounted for in both the experimental design and analysis, these agents function as invaluable analogs with which to probe the altered states of arousal of coma and slow-wave sleep [3,20].

### 3.2. Anesthesia and Sleep

Anesthesia shares a number of mechanistic overlaps with sleep. First and foremost, anesthesia co-opts endogenous sleep-related circuits by recruiting sleep-promoting nuclei such as the lateral habenula and the ventrolateral preoptic area, and by simultaneously suppressing arousal-promoting nuclei such as the tuberomammillary nucleus and the dorsal raphe [9,45,46]. More recently, it has been demonstrated that the optogenetic reactivation of a population of neuroendocrine anesthesia-activated neurons in the supraoptic nucleus promoted slow-wave sleep, and that ablation of these same cells conversely led to a significant loss of both slow-wave and REM sleep, and a shorter duration of general anesthesia [47]. Furthermore, sleep and anesthesia have a reciprocal influence on one another; sleep deprivation affects both the induction and recovery from anesthesia [48], and sleep debt induced by sleep deprivation can be attenuated by specific anesthetics [49,50]. These commonalities imply that there are shared neurobiological processes across the two conditions. As such, while anesthesia may not be a perfect replication of physiological sleep [51], the many mechanistic and behavioral overlaps between these two states of unconsciousness, combined with the comprehensive experimental control granted by anesthesia make it an optimal tool for the unravelling the intricacies of the spontaneous EEG dynamics of sleep.

Of all the anesthetics we tested, at a consistent and deep surgical plane of anesthesia, only urethane and CH produced the spontaneous, cyclic alternations between a REM-like activated state and an nREM-like deactivated state consistent with the quintessential EEG features and dynamics observed in natural sleep. Urethane has long been recognized as an unusual anesthetic in terms of the EEG patterns it produces [52,53,54], and its primary mechanism for inducing unconsciousness, which is to hyperpolarize central-nervous-system neurons by potentiating resting-potassium conductance, thus decreasing membrane input resistance [18]. Moreover, the exceptionally slow pharmacokinetics of urethane induces a long lasting, stable plane of surgical anesthesia with minimal depression of both the autonomic nervous system and neurotransmission in the central nervous system [34,55,56]. Unfortunately, this prolonged metabolism of urethane also contributes to the sustained exposure to its carcinogenic effects [19,55]. Due to these ethical considerations, urethane anesthesia is typically limited to acute animal experimental preparations [55].

Consequently, it is of great interest that CH anesthesia produced a pattern of EEG activity remarkably similar to urethane, as CH is considered an acceptable anesthetic for recovery surgeries [57,58], and indeed is still in use as a clinical sedative primarily for pediatric patients [59]. This suggests that CH could provide researchers an avenue to perform controlled neurophysiological manipulations under a sleep-like state of CH anesthesia, and then assess behavioral outcomes. Such a paradigm would be especially useful in assessing the role of brain state in memory consolidation and may also provide an ethical alternative to methods that are currently employed to bias brain state, such as sleep deprivation, which inevitably induces stress. Yet, the extent to which CH mimics the dynamic physiological measures observed in both urethane and sleep remains unknown, and may not be identical, as it has been reported to dose-dependently depress cardiovascular and respiratory functions [57]. Nonetheless, there are several pharmacological similarities between CH and urethane that suggest a greater overlap in neurophysiology may exist.

### 3.3. Pharmacological Mechanisms Influencing Brain State

We theorize two potential reasons for the inconsistencies in brain state evoked by CH and the other GABAergic anesthetics. First, at the dosages required to produce a surgical plane of anesthesia, all of the anesthetic agents we tested are acting on multiple neurophysiological targets in addition to potentiating GABAergic activity [19,46,60]. Secondly, CH is unique from the other GABAergic anesthetics, as it is first metabolized into 2,2,2-trichloroethanol, which then exerts the anesthetic effects that potentiate GABA-A-receptor-mediated activity [46,61,62]. Interestingly, urethane is also metabolized into ethanol and carbamic acid [55] (albeit much more slowly [63,64]). Both ethanol and urethane are considered to have a diffuse spectrum of action, with actions that modestly enhance activity on glycine, GABA-A and nicotinic acetylcholine receptors, while simultaneously inhibiting AMPA and NMDA receptors [19,65]. Subsequently, the metabolism of CH into a chlorinated ethanol isomer may partially elucidate the parallels in EEG activity and dynamics we observed during both CH and urethane anesthesia, in addition to the inconsistencies with the pattern of burst-suppression activity evoked by other GABAergic anesthetics.

## 4. Materials and Methods

### 4.1. Subjects

Thirty-six naïve male Sprague-Dawley rats (Charles River, and University of Alberta Science Animal Support Services) weighing on average 287.4 ± 8.5 g (mean ± SEM) were randomly assigned to one of the following anesthetic groups for acute electrophysiological recordings: urethane (*n* = 10), KET-XYL (*n* = 5), ISO (*n* = 5), PTB (*n* = 6), PRO (*n* = 5), or CH (*n* = 5). Animals were kept on a 12 h light/dark cycle at 20 ± 1 °C and housed in cages with no more than 4 rats per cage. All surgical procedures outlined herein conform to our animal-use protocol (092) that was approved by the Biological Sciences Animal Care and Use Committee of the University of Alberta, in accordance with the guidelines of the Canadian Council on Animal Care.

### 4.2. Surgery and Anesthesia

Initial induction of animals occurred in a plexiglass anesthetic chamber using 4% ISO in medical (100%) oxygen. Upon the loss of righting reflexes [66], animals were transferred to a nose cone and maintained at 1.5–2.5% ISO. All rats, except those assigned to the ISO-anesthetic group, were subsequently implanted with a jugular catheter to allow for intravenous (i.v.) administration of their assigned anesthetic. Including knock-down time, the procedure to implant the jugular catheter took approximately 10–12 min, so rats in drug groups other than ISO received only a short exposure to ISO. Rats in the ISO-anesthetic group were continuously administered anesthesia in gaseous form via a nose cone and were maintained at a surgical plane of anesthesia using 2% ISO, unless manipulated to evaluate changes in EEG dynamics associated with anesthetic depth. For all other animals, ISO was immediately discontinued following the implantation of the jugular catheter, and they were switched to i.v. administration of their respective anesthetic. Additionally, stereotaxic and surgical procedures outlined below took a minimum of 1 h, so animals not in the ISO group had ample time to exhale and metabolize any excess ISO prior to EEG recordings.

To ensure animals were maintained at a surgical level of anesthesia, changes in anesthetic state were continuously monitored by observing for any changes in heart or breathing rates, particularly when changes in brain state were observed, and subsequently verified by administering a hind-paw pinch. If a reflexive withdrawal was observed, then supplemental bolus doses (2% of the original dose) of the assigned anesthetic were administered until a stable plane of surgical anesthesia was restored. The dosages used to establish a surgical plane for each i.v. anesthetic group were as follows: urethane (1.70 g/kg); KET-XYL (93.8 mg/kg, 9.24 mg/kg); PTB (65.0 mg/kg); PRO (8.00 mg/kg bolus, 60.0 mg·kg^−1^·h^−1^ continuous infusion); CH (200.0 mg/kg bolus, 150.0 mg·kg^−1^·h^−1^ continuous infusion). Additional bolus doses of anesthetic were only delivered either when evaluating the effects of an increased dose of anesthetic on ongoing EEG measures, or if the animal exhibited a loss of surgical plane as evidenced by a reflexive withdrawal to a hind-paw pinch.

### 4.3. Stereotaxic and Recording Procedures

Following the initial surgical procedures, rats were secured in a stereotaxic frame (Kopf Instruments, Tujunga, CA, USA). Core body temperature was monitored and maintained at 37 °C for the duration of the experiment using a servo-driven system connected to a heating pad and rectal probe (TR-100, Fine Sciences Tools, Vancouver, BC, Canada). For three of the drug groups (ISO, PRO, and CH), rats received continuous delivery of the anesthetic. This was achieved using a modified nose cone attached to the stereotaxic frame in the ISO-anesthetic group, and a continuous infusion pump (Harvard Apparatus, Holliston, MA, USA) in the PRO and CH anesthetic groups.

Teflon-coated stainless-steel wire was used to construct all recording electrodes (bare diameter 125 µm; A-M Systems Inc., Sequim, WA), and placement of recording electrodes was conducted using stereotaxic coordinates measured in relation to bregma. These electrodes were placed in the frontal neocortex (AP: +2.8 mm; ML: ± 2.0 mm) in either superficial (DV: −0.1 to −0.5 mm) or deep layers (DV: −1.0 to −1.3 mm), and the hippocampus (AP: −3.3 to −3.5 mm; ML: ± 2.3 to 2.5 mm; DV: −2.4 to −3.3 mm) to confirm any observed changes in forebrain state.

Based on previous work from our lab [22], recorded field-potential signals from the cortex and hippocampus were either: (1) measured against an electrically neutral reference point, which was typically the grounded stereotaxic frame, but in a few cases was a low-resistance uninsulated Teflon wire ~2mm long placed vertically through the layers of frontal cortex, or (2) were differentially amplified by referencing one tip of a staggered bipolar electrode to another. Field-potential recordings were then amplified at a gain of 1000, and then filtered between 0.1 to 500 Hz using a differential AC amplifier (Model 1700, A–M Systems Inc.; Sequim, WA, USA). Recorded signals were digitized online (at sampling frequency of 1 kHz) using either a Digidata 1322A A–D board in conjunction with the acquisition program AxoScope (Axon Instruments; Union City, CA, USA), or a PowerLab Pro combined with Lab Chart 8 software (AD Instruments; Colorado Springs, CO, USA).

### 4.4. Experimental Design

Cortical and hippocampal field-potential activity was recorded for a minimum of 70 min for all animals regardless of anesthetic group. For the purposes of this study, a surgical plane of anesthesia was operationally defined as a loss of reflexive withdrawal from a nociceptive stimulus (i.e., withdrawal of a paw when pinched) and an absence of reactivity while in the stereotaxic frame. All animals were maintained at surgical plane of anesthesia at the average effective dosage of the designated anesthetic. Changes in electrophysiological activity in response to discontinuation of supplementary doses of anesthesia were recorded for 10–40 min depending on when the animal responded to the nociceptive stimulus. Subsequently, the animal was restored to a surgical plane using supplementary doses of anesthetic. Following termination of the EEG recording, rats were euthanized by transcardial perfusion with saline.

### 4.5. Statistical Analyses

Analyses were conducted offline on the previously acquired digitized files. Visual inspection of raw EEG signals was used to segment data based on recorded bolus i.v. administrations of anesthetics or tagged reports of changes in anesthetic state. Data were segmented into two groups: 40 min epochs at a given anesthetic dose, to characterize changes in spectral power over time using a spectrogram, and 2–5 min epochs to assess differences in spectral power at specific time points in the recording. Spectral power was computed using Welch’s periodogram method on Hanning-windowed data of 6 s epochs with a 2 s overlap (MATLAB: Mathworks, Natick, MA, USA) for analysis. Spectrograms were computed using 30 s epochs separated by 10 s across the analyzed segment. The computation of spectral power also included an estimate of the upper and lower 95% confidence limits, which enabled us to calculate significant changes in power across our manipulations. Plots of the analyzed data were subsequently created using Origin software (Microcal Software, North Hampton, MA, USA). Averages were computed as arithmetic means and include the standard error of the mean.

Both state changes and cycle durations in urethane and CH were characterized by first extracting the peak frequency of slow-oscillatory power in the cortex, either alone or along with the peak frequency of theta in the hippocampus, to create a ratio of slow-oscillatory power compared to theta power over time. Then, the state-change threshold was established by determining the saddle point of the bimodal power distribution, allowing for the calculation of cycle duration and the proportion of time spent in deactivated or activated states on a cycle-by-cycle basis. In CH, where rats spent 100% of time in deactivated patterns following bolus doses, and no threshold crosses occurred, cycles were estimated based on any small changes observed in the raw EEG traces or spectrograms.

Changes in the duration of inter-burst intervals during burst-suppression activity in the PTB, ISO and PRO groups were compared over a span of 3 min both pre- and post-bolus administration for each anesthetic using a 2-way ANOVA. Since we were only interested in the effect of dose, only this component of the analysis was reported. The threshold for detection of burst activity was set at half of the peak amplitude over the six minutes assessed per animal, with an average threshold of 0.18 ± 0.01 mV (*n* = 9). Statistical significance was set at α = 0.05. All data will be made publicly available on the Dataverse repository, accession numbers will be provided during the review process.

## 5. Conclusions

While the concept of agent-specific effects on brain state such as those we observed in the current study may be considered common knowledge to many researchers in the field of anesthesia, often the effects of a chosen anesthetic on brain state and the related evaluations can be overlooked when conducting neurophysiological experiments [10]. Our results clearly demonstrate why the choice of anesthetic agent is a crucial element for researchers to take into consideration when designing and conducting neurophysiological experiments under anesthesia, as the evoked brain state can significantly impact the experimental output. Furthermore, as EEG becomes a more prevalent tool for clinically measuring the depth of unconsciousness in both anesthesia and coma, it will be paramount to explore the pharmacological mechanisms of action driving agent-specific differences in EEG signatures between anesthetics [46], in order to improve clinical perioperative outcomes [67].

A better understanding of the intricacies of both the neural targets and pathways that interplay to induce and maintain unconsciousness in anesthesia not only has implications for the refinement and optimization of anesthetic interventions, but also for understanding the neurobiological causes of sleep disorders [68]. The overlap with natural sleep of the electrophysiological dynamics observed in CH, in addition to the physiological overlaps observed in urethane suggest that these anesthetics are unprecedented tools for probing unconsciousness and the extent to which anesthesia co-opts endogenous sleep pathways. This is because the intertwined pharmacological and physiological unconscious states of sleep and anesthesia are mutually informative, and ultimately offer novel opportunities to explore the mechanisms responsible for unconsciousness [1].

## Figures and Tables

**Figure 1 ijms-23-03608-f001:**
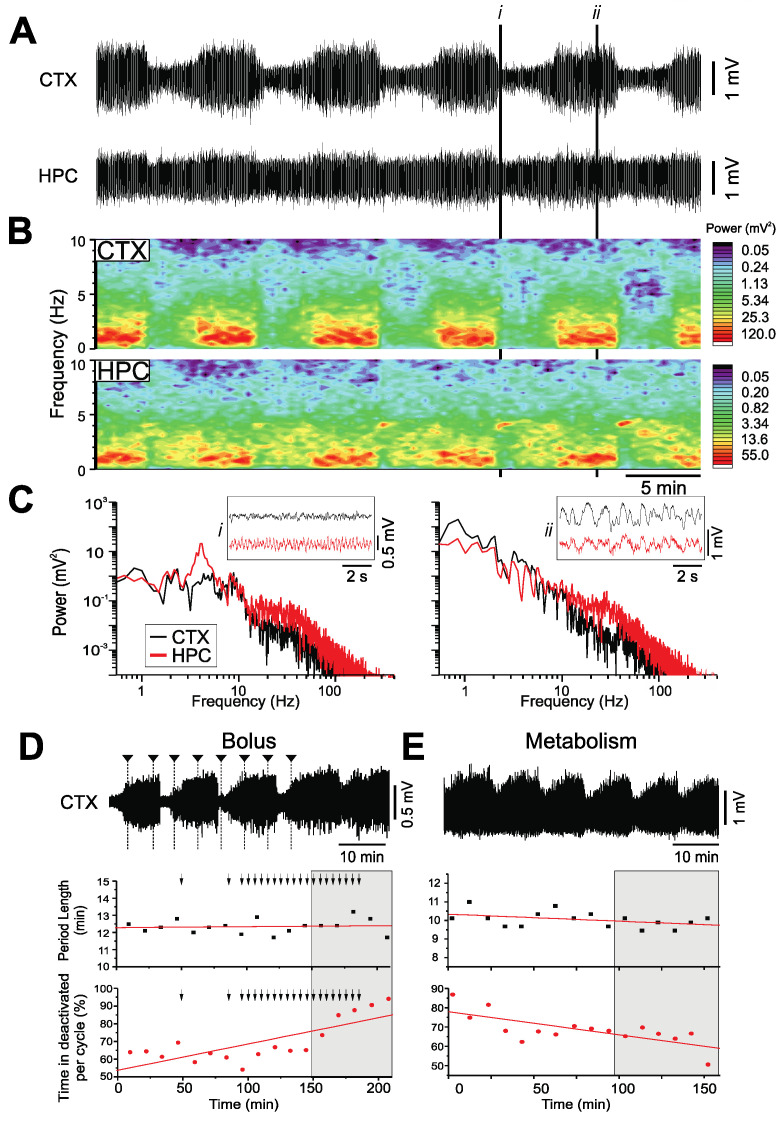
Spontaneous and cyclic alternations of brain state under urethane anesthesia. (**A**): Continuous, 40 min duration cortical (CTX) and hippocampal (HPC) EEG traces. 10 s samples of activated (i) and deactivated (ii) states are expanded in C. (**B**): Spectrograms of CTX (top) and HPC (bottom) EEG traces in A. (**C**): Power spectra for the CTX and HPC during an activated state (left), and a deactivated state (right), with insets of 10 s raw traces representative of each state. (**D**): 60 min cortical EEG sample (top). Arrows indicate administration of supplemental doses of urethane in increments of 0.01 mL. The sample trace is denoted by the grey box in both the scatterplots of the period length of cycles across time (linear fit, *n* = 17, *p* = 0.74) and the percentage of time spent in deactivated per cycle (linear fit, *n* = 17, *p* < 0.01). (**E**): 60 min cortical EEG sample (top) of metabolism of urethane over time. The sample trace is denoted by the grey box in both the scatterplots of the period length of cycles across time (linear fit, *n* = 16, *p* = 0.11) and the percentage of time spent in deactivated per cycle (linear fit, *n* = 16, *p* < 0.01).

**Figure 2 ijms-23-03608-f002:**
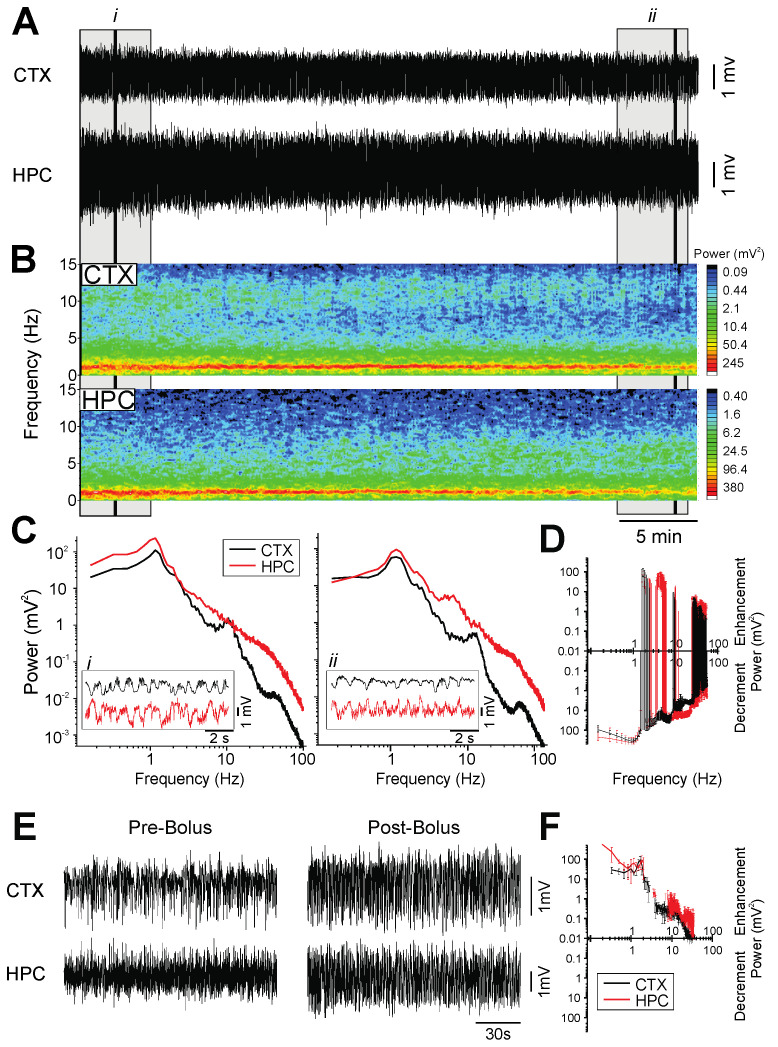
Global slow-oscillatory activity under ketamine-xylazine anesthesia. (**A**): Continuous, 40 min duration cortical (CTX) and hippocampal (HPC) EEG traces. 5 min (grey box) and 10 s (black line) samples from the beginning (i) and end (ii) of the 40 min recording are further analyzed in in C. (**B**): Spectrograms of CTX and HPC traces from A. (**C**): Power spectra of a 5 min selection (grey box) from the EEG traces in A, with an inset of a 10 s raw CTX and HPC EEG trace (black line) from the beginning of A (left panel, i) and end of A (right panel, ii). (**D**): The average difference in spectral power over a 40 min metabolism period, plotted using 2 min samples pre- and post-metabolism to denote the average increment or decrement of power (*n* = 3). (**E**): *Pre-bolus:* A 2 min sample of CTX and HPC traces during a surgical plane of ketamine-xylazine anesthesia. *Post*-*bolus*: A 2 min sample of CTX and HPC traces following a bolus infusion of ketamine-xylazine (16.6 ± 1.9 mg/kg and 1.5 ± 0.1 mg/kg; *n* = 5). (**F**): The average difference in spectral power following bolus infusion, plotted using 2 min samples pre- and post-bolus to denote average increment or decrement of power (*n* = 5).

**Figure 3 ijms-23-03608-f003:**
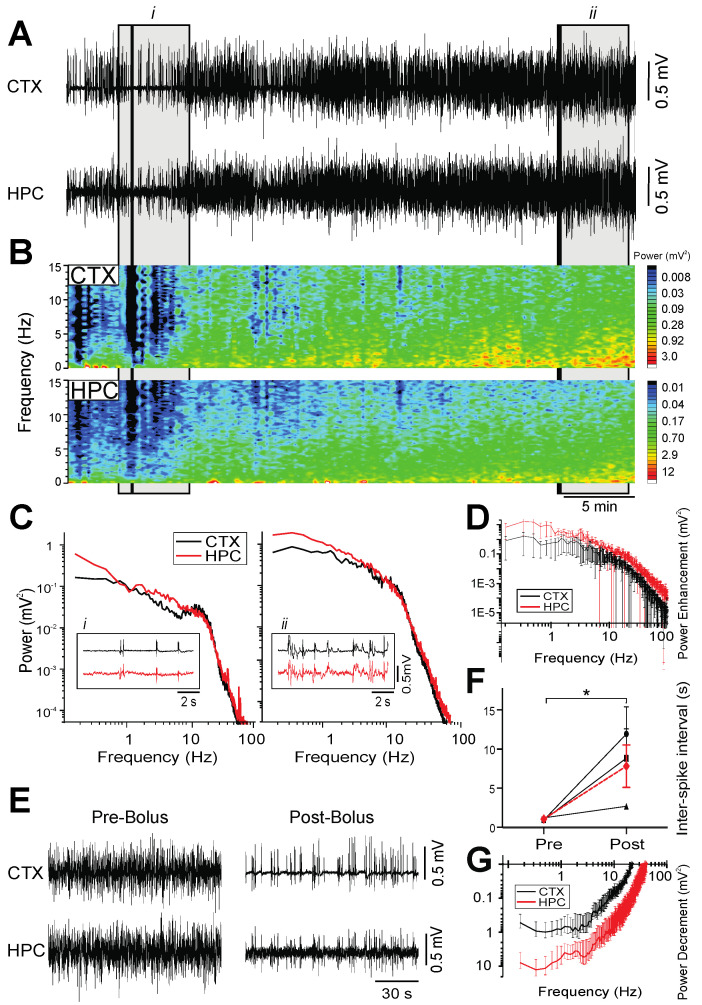
Burst-suppression activity under pentobarbital anesthesia. (**A**): Continuous, 40 min duration cortical (CTX) and hippocampal (HPC) EEG traces. 5 min (grey box) and 10 s (black line) samples from the beginning (i) and end (ii) of the 40 min recording are further analyzed in in C. (**B**): Spectrograms of CTX and HPC EEG traces in A. (**C**): Power spectra of a 5 min selection (grey box) from the EEG traces in A, with an inset of a 10 s raw CTX and HPC EEG trace (black line) from the beginning of A (left panel, i) and end of A (right panel, ii). (**D**): The average difference in spectral power over a 40 min metabolism period, plotted using 2 min samples pre- and post-metabolism to denote the average increment or decrement of power (*n* = 3). (**E**): Pre-bolus: A 2 min sample of raw CTX and HPC traces during surgical plane of pentobarbital anesthesia. Post-bolus: A 2-min sample of activity following a bolus infusion of pentobarbital (6.5 mg). (**F**): The duration of isoelectricity (ISI) in pentobarbital-anesthetized rats significantly increased by 6.8 ± 2.8 s (*n* = 3; F_(1,503)_ = 163.5; *, *p* < 0.0001) following a 6.5 mg bolus dose of pentobarbital (individual animals are represented in black, average in red). (**G**): The average difference in spectral power pre- to post-bolus infusion, plotted using 2 min samples pre- and post-metabolism to denote increment or decrement of power (*n* = 3).

**Figure 4 ijms-23-03608-f004:**
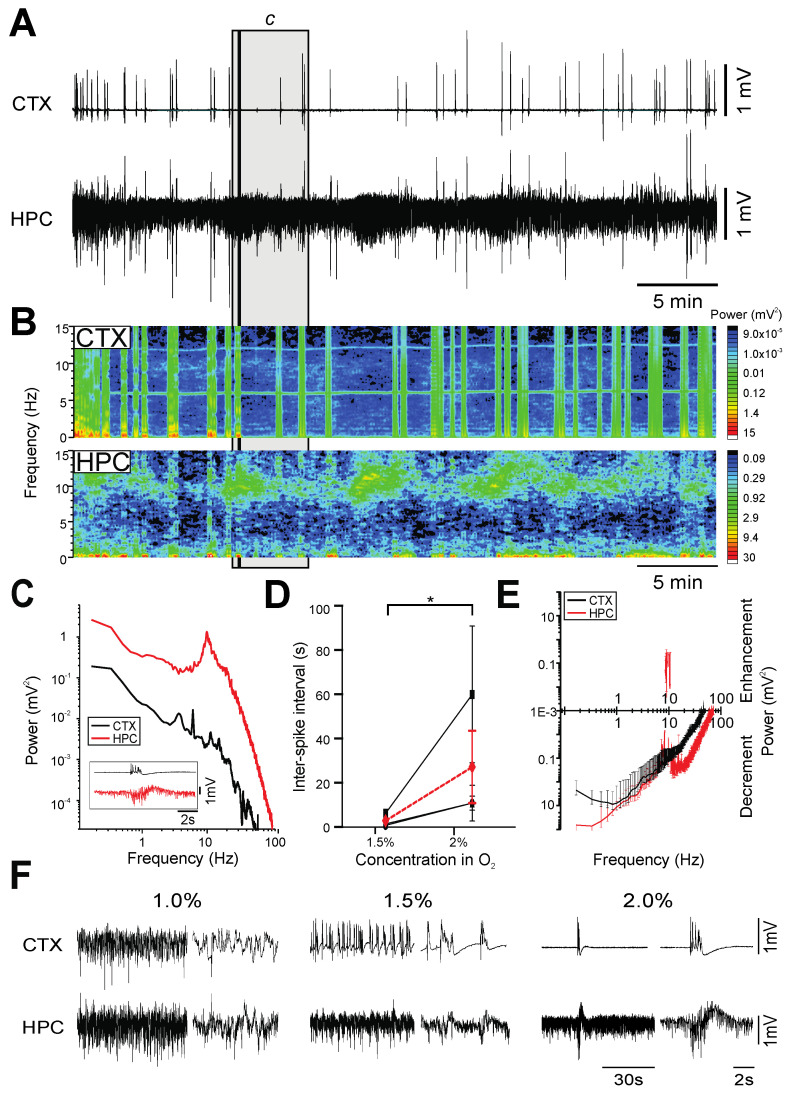
Burst-suppression activity under isoflurane anesthesia at 2%. (**A**): Continuous, 40 min duration cortical (CTX) and hippocampal (HPC) EEG traces. A 5 min (grey box) and 10 s (black line) sample from 40 min at 2% isoflurane from the 40 min recording are further analyzed in in C. (**B**): Spectrograms of CTX and HPC EEG traces in A. (**C**): Power spectra of a 5 min selection (grey box) from the EEG traces in A, with an inset of a 10 s raw CTX and HPC EEG trace (black line) from A. (**D**): The duration of isoelectricity (ISI) in isoflurane-anesthetized rats significantly increased by 24.4 ± 14.6 s (*n* = 3; F_(1,263)_ = 96.11; *, *p* < 0.0001) following a shift from 1.5% to 2% isoflurane (individual animals are represented in black, average in red). (**E**): The average differences in spectral power between 2 min samples from 1.5% and 2% isoflurane are plotted to denote average increment or decrement of power with increased isoflurane concentration (*n* = 3). (**F**): 1 min samples of CTX and HPC EEG traces at 1%, 1.5% and 2%, with 8 s expansions to the right of each 1 min trace.

**Figure 5 ijms-23-03608-f005:**
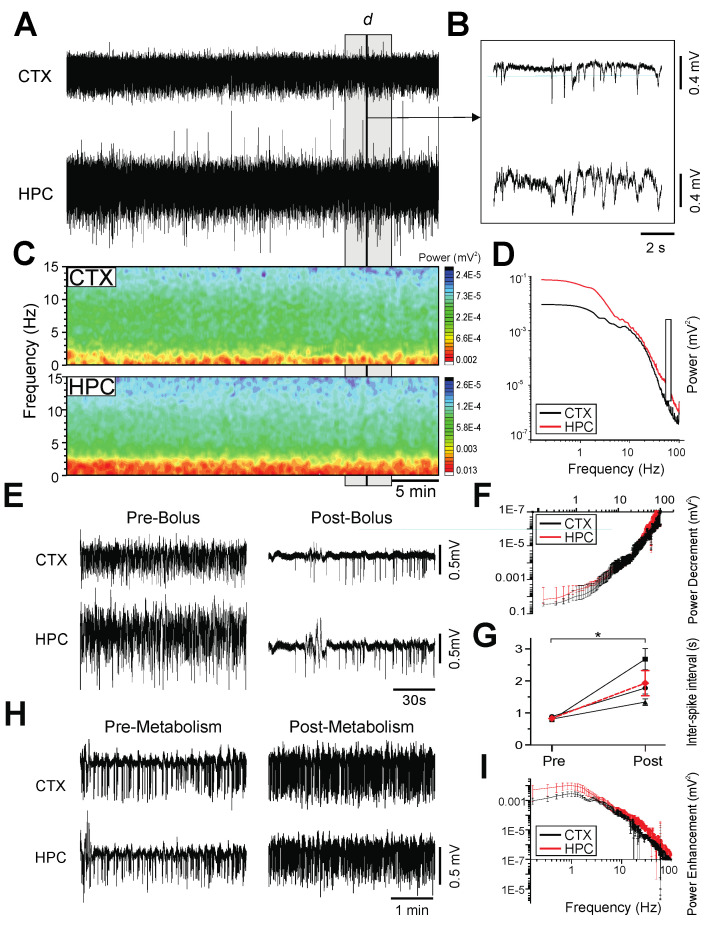
Burst-suppression activity under propofol anesthesia. (**A**): Continuous, 40 min duration cortical (CTX) and hippocampal (HPC) EEG traces. A 5 min (grey box) sample is further analyzed in D, and a 10 s (black line) sample is expanded in B. (**B**): Expanded 10 s trace of CTX and HPC from A (black line). (**C**): Spectrograms of CTX and HPC traces in A. (**D**): Power spectra of a 5 min selection from A (grey box), representative of the activity observed throughout the 40 min duration of the trace in A, due to continuous delivery of propofol. Noise at 60 Hz has been obfuscated by a white box. (**E**): Pre-bolus: A 2 min sample of CTX and HPC traces during a surgical plane of propofol anesthesia. Post-bolus: A 2 min sample of activity following a bolus infusion of propofol (2 mg). (**F**): The average difference in spectral power following a bolus (2 mg) of propofol, plotted using 2 min samples pre- and post-metabolism to denote the average increment or decrement of power (*n* = 3). (**G**): Inter-spike intervals in propofol-anesthetized rats significantly increased by 1.1 ± 0.4 s (F_(1,574)_ = 136.0; *, *p* < 0.0001; *n* = 3) following a 2 mg bolus dose of propofol. (**H**): Pre-metabolism: A 4 min sample of pre-metabolism activity in the CTX and the HPC during propofol anesthesia. Post-metabolism: A 4 min sample of activity following 10 min metabolism of a bolus. (**I**): The average difference in spectral power over a 10 min metabolism period, plotted using 2 min samples pre- and post-metabolism to denote the average increment or decrement of power (*n* = 3).

**Figure 6 ijms-23-03608-f006:**
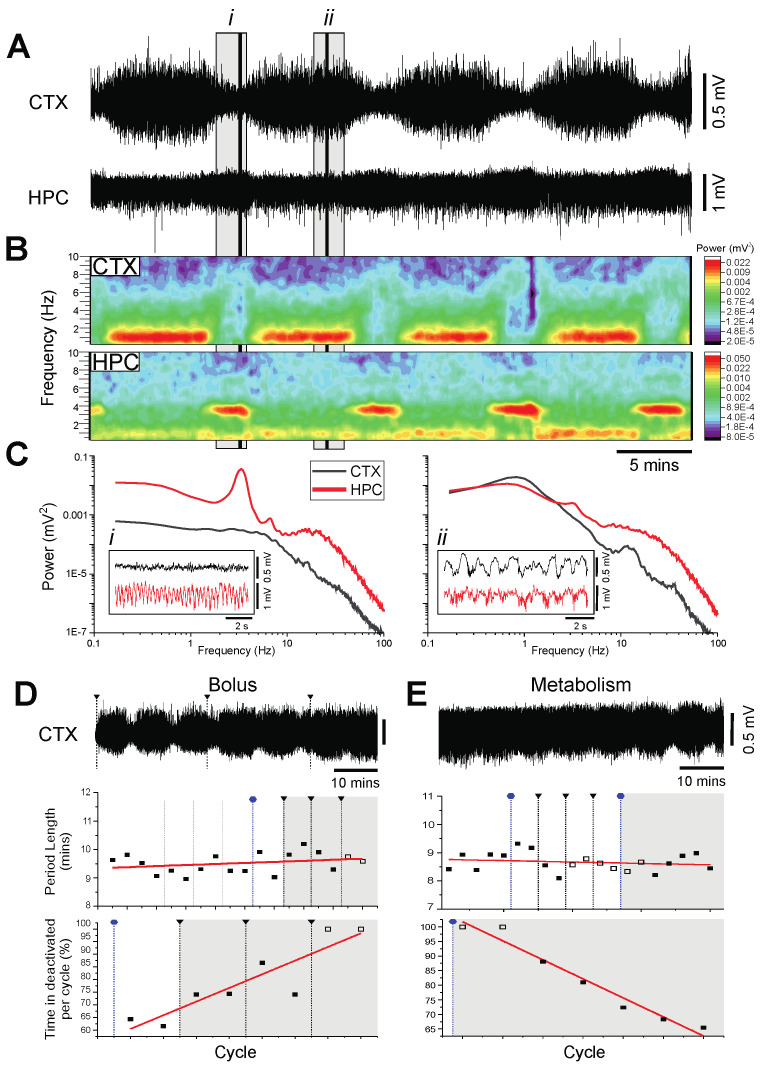
Spontaneous and cyclic alternations of brain state under chloral-hydrate anesthesia. (**A**): Continuous, 40 min duration cortical (CTX) and hippocampal (HPC) EEG traces. 10 s samples of activated (i) and deactivated (ii) states are further analyzed in C. (**B**): Spectrograms of CTX (top) and HPC (bottom) EEG traces in A. (**C**): Power spectra for the CTX and HPC during an activated state (left, i), and a deactivated state (right, ii), with insets of 10 s raw traces representative of each state. (**D**): 65 min cortical EEG sample (top). Black lines indicate administration of supplemental doses of chloral hydrate in increments of 15 mg (0.15 mL). The sample trace is denoted by the grey box in both the scatterplots of the period length of cycles across time (linear fit, *n* = 18, *p* = 0.26) and the percentage of time spent in deactivated per cycle (linear fit, *n* = 8, *p* = 0.003). Grey lines indicate supplemental doses of 5 mg of chloral hydrate (0.5 mL), and blue lines indicate a stoppage of continuous infusion of chloral hydrate. Continuous infusion resumed with the next black line and bolus infusion. Unfilled boxes are estimated cycles (see methods). (**E**): 65 min cortical EEG sample (top) of metabolism of chloral hydrate over time. The sample trace is denoted by the grey box in both the scatterplots of the period length of cycles across time (linear fit, *n* = 20, *p* = 0.44) and the percentage of time spent in deactivated per cycle (linear fit, *n* = 7, *p* < 0.001). Blue lines indicate a stoppage of continuous infusion of chloral hydrate. Continuous infusion resumed with the next black line and bolus infusion. Unfilled boxes are estimated cycles (see methods).

## Data Availability

All data will be made publicly available on the University of Alberta Dataverse repository, accession numbers will be provided during the review process.

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
