# Peer review of "A Comparison of Brain-State Dynamics across Common Anesthetic Agents in Male Sprague-Dawley Rats"

_ijms, 2022, doi:10.3390/ijms23073608_

Round 1

Reviewer 1 Report

The paper by Flannegan and co-workers describes the electrophysiological (EEG) features associated with the use of six different anesthetic agents. The paper is well written, but the data are nor novel at all. Moreover, the paper has two major concerns:

  • first of all, the significance of oscillatory patterns are different between cortical and subcortical recordings (i.e. burst suppression patterns likely have a different functional relevance between cortex and sub-cortical structures).
  • The translational impact of the research is very low. In particular, the Authors made a series of mistakes about the interpretation of different EEG patterns in human diseases. For istance, it is well known that every sedation may induce every kind of EEG changes. The presence of a suppression pattern, rather than a burst suppression pattern, is not specific for a drug and does not allow to differentiate among distinct prognostic patterns, neither among different clinical conditions. This interpretation (I think also in animals ...) relies on  different features, not assessed in the present paper, including the shape of ictal and inter-ictal activity and, possibly, also non-linear EEG analyses.

Reviewer 2 Report

The manuscript evaluates the effect of different anesthetic agents that are widely used in research on the brain’s global network activity and tries to describe the different brain states induced by the different drugs. The authors demonstrate that certain agents tend to cause the brain to switch to a burst suppression mode, which is very different from sleep, while other drugs create a different, more “sleep like”, oscillatory states.

The work is well done, and the data presented clearly. The results are of good quality and worthy of publication. However – the authors tend to overstate the results and draw conclusions that are somewhat beyond what they actually demonstrate. The work could benefit from presentation of further analysis of the data as detailed below.

Comments:

- The authors claim that isoflurane, propofol and pentobarbital are effective for the study of burst suppression only, despite the fact that both propofol and isoflurane allow titration over a wide range: from deep anesthesia to light sedation, including intermediate anesthetic plan that presents as a slow oscillatory pattern. Surprisingly, the authors could not find a lower dose that allowed complete unresponsiveness. The reason that they got such different results with chloral hydrate compared to the other GABA agonists could be that this drug is less potent, which allowed them to find the “sweet spot” where the animal is unresponsive but the EEG is oscillatory. In any case, even if we accept the claim that such point is difficult to find, note that animals under natural sleep also withdraw from pain. Therefore, if any conclusion can be made it’s that these drugs are a better model of sleep than Urethane and chloral hydrate.

- In the clinical world, as well as in modern research, these drugs are used in combination with analgesic drugs, which suppress the response to painful stimuli and allows working at much lower doses. Thus, many of the comparisons made here (especially those made under burst suppression conditions) are invalid.

- The authors should be careful about the claims and statements they make regarding which drug is useful to model which brain state – the drugs act on multiple targets that are different from the condition they try to model. The drugs may create different states at different doses. It is important that researchers know the effects of the drugs they use in their studies on the brain state, and in this respect, the current work is an important statement. However, it would be a mistake to consider a certain drug as the optimal model for sleep, head injury or other condition based on the EEG pattern at a specific area under a single dose of the drug.

- The authors recorded simultaneously from two regions, but refrain from mentioning the relations of the activity between the regions (or measuring the coherence, phase or any other measure of correlated activity). This is most obvious when comparing urethane and chloral hydrate anesthesia, where it seems that for the first, the cortex is in sync with the hippocampus, while for the second they’re at 180 degree to each other. Would it be possible to relate to this measure?

- Many anesthetic drugs were reported to cause oscillations at much higher frequencies (for example – gamma oscillations under ketamine) why did the authors limit their investigation to frequencies below 15 Hz?

- All the figures are examples of single cases. No summed data is presented. Any reason for this odd presentation?

- Why did the author choose Ket-xyl combination rather than examine the effect of single drugs like with all the other drugs? This is problematic since the effect of xyl is longer than that of ket. As a result, the change following bolus administration reflects mostly the change in ketamine effect. Whereas during “metabolism” they see mostly the effects of xyl.

- In figure 2, it seems that ketamine creates cortical (but not hippocampal) alpha oscillations. Was this a constant feature?

- The authors present a very simplistic approach to the molecular targets of the different anesthetics (see introduction lines 49-51). A through detailed description is beyond the scope of this paper, but as they try to compare the effects of this drug (which are mostly “dirty” - effecting more than a single target). They should at least relate to the fact that this is a very crude and general approximation.

Author Response

Please see attachment. Thank you for your feedback!

Reviewer 3 Report

Reviewer’s comment

 The anesthetic effects could be different depending on dosages, animal species used, and the pharmacological mechanisms specific to each anesthetic agent. In this study, the authors conducted a controlled comparison of spontaneous electrophysiological dynamics at a surgical plane of anesthesia under six common research anesthetics, using a ubiquitous animal model. This is a interesting research topic. However, there are some major issues with methodology and data interpretation that need to be clarified.

Major comments

  1. According to the methods sections, all rats except the isoflurane group received isoflurane administration prior to IV jugular cannulation. In addition, all groups underwent stereotactic frame application surgery for EEG recording under isoflurane anesthesia prior to drugs administration through IV catheter. So, I wonder if the residual isoflurane effect and surgical stimuli before drug’s administration could affect the EEG results of each group. Why did the authors not perform surgical procedures such as cannulation and EEG electrode insertion on a day other than same time?
  2. My main concern is that the effects of anesthetics do not remain stable throughout the experiment for all groups. After a single dose of drugs, there are peaks and troughs. Due to inter-subject variability of pharmacokinetic and pharmacodynamic parameters, anesthetic effects can differ from rat to rat, even though the same dose was administered to rats, resulting in different EEG patterns associated with various levels of anesthesia. How did authors estimate that the surgical plane of anesthesia was consistent across the rats?
  3. How was the supplemental dose and frequency of drugs (Urethane, Ket-Xyl.., etc.) determined for additional administration? If additional doses of its dugs are frequently administered, there may be significant alteration of its anesthetic effects or changes of surgical plane of anesthesia aside from the changes of anesthetic effects of the first administered drug over time
  4. The dosage of the drug appears to be higher than the recommended dosage in clinical practice. For example, for CH, the oral or IP route is preferred over the IV route. Also, the recommended dose is less than the dose used in this study. (about less than the half of the used dose in this study) So, how did you determine the dosage of each drugs in this study? Also, If this dose is higher than the recommended dose, could a deeper level of anesthesia affect the outcome of this study?

Minor issues

  1. (Fig 1E, Fig 2D, Fig.5H, Fi.6E) I am not sure of the meaning of metabolism. What does metabolism indicate? Does it mean that it takes a long time after applying drugs? How did you quantify the metabolism of anesthetics? Is there a possibility of inter-subject variability in terms of drug distribution and elimination across the rats?
  2. (Fig 2,3,4,6 C) The raw EEG is inserted into the low part of the spectrogram. This confuses the interpretation of Figures. It is recommended to draw the raw EEG on an insert similar to Figure 1C.
  3. Although there is a figure legend for figure 3H, figure 3H does not exist. Figure 3H Legend is shown in Figure 3G. There is even no mention of Figure 3H in the results section.
  4. (Fig. 2D and F, 3D and G 4E, 5F and I) I don’t understand what Figures means? What does the y-axis mean? Do the numbers on the y-axis represent the percentage of change? Can you elaborate on the meaning of the numbers on the y-axis?
  5. What is the shallow rectangular box in the spectrogram? (Fig. 5D)

Author Response

(The authors gave the same response as above.)

Round 2

Reviewer 3 Report

The points raised by the reviewer was clearly addressed. Thank you for your hard work. I hope that this paper will serve as a reference for neurophysiological experiments.